# The Effects of Different Feeding Regimes on Body Composition, Gut Microbial Population, and Susceptibility to Pathogenic Infection in Largemouth Bass

**DOI:** 10.3390/microorganisms11051356

**Published:** 2023-05-22

**Authors:** Yao Zheng, Haojun Zhu, Quanjie Li, Gangchun Xu

**Affiliations:** Key Laboratory of Integrated Rice-Fish Farming Ecology, Ministry of Agriculture and Rural Affairs, Freshwater Fisheries Research Center (FFRC), Chinese Academy of Fishery Sciences (CAFS), Wuxi 214081, China; zhengy@ffrc.cn (Y.Z.); zhuhj@ffrc.cn (H.Z.); liqj@ffrc.cn (Q.L.)

**Keywords:** iced fish, culture mode, pathogenic source, intestinal microbiota, metagenomics

## Abstract

This study investigated the effects of dietary commercial feed (*n* = 50,025 in triplicate, named group PF for soil dike pond, sampling *n* = 7; *n* = 15,000 in triplicate, WF for water tank, *n* = 8), iced fish (*n* = 50,025 in triplicate, PI, *n* = 7), and a combination of both (*n* = 50,025 in triplicate, PFI, *n* = 8) on different metabolic parameters of the largemouth bass, *Micropterus salmoides* (0.67 ± 0.09 g, culture period from June 2017 to July 2018). Throughout the experimental period, different areas of water (including input water of the front, middle of the pond, and from the drain off at the back) and their mixed samples were simultaneously analyzed to find the source of the main infectious bacteria. Various feeding strategies may differentially affect body composition and shape the gut microbiota, but the mode of action has not been determined. Results showed that no significant differences were found in the growth performance except for the product yield using a different culture mode (PFI vs. WF). For muscle composition, the higher ∑SFA, ∑MUFA, ∑n-6PUFA, and 18:3n-3/18:2n-6 levels were detected in largemouth bass fed with iced fish, while enrichment in ∑n-3PUFA and ∑HUFA was detected in largemouth bass fed with commercial feed. For the gut microbiota, Fusobacteria, Proteobacteria, and Firmicutes were the most dominant phyla among all the gut samples. The abundance of Firmicutes and Tenericutes significantly decreased and later increased with iced fish feeding. The relative abundance of species from the Clostridia, Mollicutes, Mycoplasmatales, and families (Clostridiaceae and Mycoplasmataceae) significantly increased in the feed plus iced fish (PFI) group relative to that in the iced fish (PI) group. Pathways of carbohydrate metabolism and the digestive system were enriched in the commercial feed group, whereas infectious bacterial disease resistance-related pathways were enriched in the iced fish group, corresponding to the higher rate of death, fatty liver disease, and frequency and duration of cyanobacteria outbreaks. Feeding with iced fish resulted in more activities in the digestive system and energy metabolism, more efficient fatty acid metabolism, had higher ∑MUFA, and simultaneously had the potential for protection against infectious bacteria from the environment through a change in intestinal microbiota in the pond of largemouth bass culturing. Finally, the difference in feed related to the digestive system may contribute to the significant microbiota branch in the fish gut, and the input and outflow of water affects the intestinal flora in the surrounding water and in the gut, which in turn affects growth and disease resistance.

## 1. Introduction

There are three feeding methods for largemouth bass (LB) *Micropterus salmoides*: iced fish used during whole-course culturing, commercial feed (occupies 10% of the total products), and their combination. In the late stage of feeding (~0.3–0.4 kg/individual), some problems, such as the reduced food intake, cause low growth rate, especially fatty liver, which still has not been substantially solved, which weakens resistance to disease. Largemouth bass are prone to congenital fatty livers because of inadequate starch and fat metabolism in the fish, which results in hepatobiliary syndrome and yellow lipid disease (adipose tissue in the muscle). It is crucial to improve the absorption and utilization rate of fat to solve the problem of the fatty liver when using commercial feed [1].

The gut microbiota is established during initial feeding and can be derived from the surrounding aquatic environment, soil/sediment, or live feed. The intestinal microbiota of fish is tightly coordinated with fish metabolism, immunity, and health in numerous fish studies [2]. Planctomycetes are dominant in the intestine of the bighead carp and are involved the glucose metabolism [3] and degradation of polysaccharides [4]. Digestibility of long-chain fatty acids in fish is lower than that of short-chain fatty acids, resulting in enhancing the population of beneficial bacteria, because the inclusion of fish waste silage improves digestive enzyme activity and nutrient digestibility. However, excessive saturated fatty acids in feed leads to a decrease in digestive absorption [5] and fatty liver, resulting from fat deposition in fish. Fatty acid synthase and NADPH supply are necessary for the liver to synthesize fatty acids, with the fatty acid composition of feed having a certain regulatory effect on β-oxidation [6]. Its composition in fish bodies and tissues can closely reflect the fatty acid composition of the feed [7].

Largemouth bass have elongated intestines and sacculated caeca that facilitate microbial fermentation, which is beneficial for a higher growth rate [1]. These properties make it a great fish candidate for intensive aquaculture in many countries, such as China. Problems including nutritional health and low feed utilization at the cultivation stage have resulted in important economic losses [8]. The different roles of the listed fatty acids and why changes in those fatty acids matter for animal health and/or for human consumption of that muscle have not been determined. Given the distinct feeding strategies [8] of different fish species, a comparison of their intestinal microbiota is important, but has not been performed [9] until now, and the cause for the higher incidence of fatty liver and disease rate remains unknown. The present study hypothesized that different feeding strategies and habitats shape the gut microbiota and muscular composition in different ways in the short- or long-term, and then possibly affect production performance. The multi-omics strategy is the best way to conduct this kind of study, which was performed in fish [2,3,9,10], and intestinal microbiota may provide novel information of the core bacterial and fungous branch that are relative to the quality of the aquatic product. Information regarding the changes in the microbiota during fish feeding strategies may contribute to advancing our understanding of the functions of the fish gut microbiota. The present study is aimed at investigating the effects of dietary administration of commercial feed and/or iced fish on various parameters, including growth performance, body composition, and gut microbiota.

## 2. Materials and Methods

### 2.1. Experimental Design and Sample Collection

Fertilized eggs of LB were obtained from the Yangzhong research center of FFRC-CAFS in autumn. The fish fry were cultivated in an indoor cement pool (20 m × 10 m × 2 m) and acclimated for 4 weeks before the experiment. The commercial feed (feed component details in Appendix A) was purchased from Jiangsu Zhe Ya Food Co., Ltd., China (Jinhua, China). Largemouth bass were fed once a day (the recommended amount was 8% of the body weight). One-month-old juvenile LB were acclimatized in a soil dike pond (9 m × 7 m × 0.2 m, 1.26 × 10^4^ L) containing dechlorinated tap water at 25 ± 1 °C, with a 14:10 h light/dark cycle. The separated filtration and recirculating system for each pond was purchased from Suzhou Municipal Shenhang Eco-Technology Development Co., Ltd. (Suzhou, China). The juvenile LB (0.67 ± 0.09 g, *n* = 500,000) were assigned to four groups distributed in 12 ponds (three ponds per group in different locations), with either 3.75 ind./m^2^ of each pond for three treatments or 136.36 ind./m^2^ of each water tank for one treatment. All the fish were equally distributed among all the four treatment groups. One-quarter of the fish were allocated to each treatment group, and data were collected from June 2017 to July 2018.

Throughout the experimental period, different areas of water (including input water of the front, middle of the pond, and back from the drain off) and their mixed samples were obtained before and after each water change. Before fish sampling, LB were anesthetized with MS-222 (150 mg/L, Sigma-Aldrich, St. Louis, MO, USA), euthanized, and intestinal content and muscle (for composition analysis) samples were collected. The microbiota for water, commercial feed, and iced fish (the mixed raw captured fish, Lianyungang Ganyu district, Jianhua aquatic products refrigeration factory) used both 16SrRNA and ITS sequences (location and group name for analysis, see Figure 1B). The three water ponds using whole-course commercial feed (5% feeding ratio and twice as the feeding rate by fish hand, 8:00/16:00) were designated as the WF group (water tank using feed; 110 m^2^, Yangzhong city, 32.297242° N, 119.80741° E). The three soil dike ponds using whole-course commercial feed were designated as the PF group (pond using feed; 13,340 m^2^, Huzhou city, 30.925088° N, 120.300707° E). The three ponds used for whole-course iced fish (Chuzhou Hengchuan Food Co., Ltd.; 8% feeding ratio and twice as the feeding rate by fish hand, 8:00/16:00) were designated as the PI group (pond using iced fish; the composition revealed in Appendix A, 13,340 m^2^, Pingwang of Suzhou city, 31.030636° E, 120.672257° N). Finally, the three ponds used for the commercial feed (8% feeding ratio for 5 months) and iced fish (8% feeding ratio and twice as the feeding rate by fish hand for 6 months, 8:00/16:00) were designated as the PFI group (13,340 m^2^, Pingwang). PF group (*n* = 7, sampled on 28 June 2018), WF group (*n* = 8, sampled on 26 June 2018), PI group (*n* = 7, sampled on 2 July 2018), and PFI group (*n* = 8, sampled on 2 July 2018). Before dissection, all specimens were surface-sterilized in 75% ethanol for 90 s and rinsed three times in sterilized distilled water. Using a stereomicroscope, fish gut dissections were carried out in a clean 90 mm diameter Petri dish using sterile phosphate-buffered-saline (PBS)-sterilized scissors and forceps. In each pond at the sampling point, three foregut contents were pooled as one individual sample and analyzed using high-throughput sequencing analysis (*n* = 3). The foregut contents were collected [10] from each fish using sterilized medical forceps and added to a tube (MO BIO Laboratories, Houston, TX, USA) using a sterile medicinal ladle. All experiments were approved by the Institutional Animal Care and Use Committee of the Ministry of FFRC-CAFS (approval ID: 2011AA1004020012).

### 2.2. Water Quality and Epidemiological Survey

Water quality was measured every 3 months (*n* = 4), and parameters analyzed were temperature (T), pH, salinity, dissolved oxygen (DO), and transparency depth (SD). Triplicate water samples from each group were transferred to a 500 mL polyethylene bottle to test physicochemical parameters of total suspended solids (TSS), total nitrogen (TN), total phosphorus (TP), and permanganate (COD_Mn_). All samples were filtered with a Whatman filter paper with a pore size of 0.45 μm before laboratory analyses. The product yield, disease happening time, death rate, fatty liver disease rate, and fat ratio were calculated based on >30 randomly selected fish samples.

### 2.3. Growth Performance and Body Composition

The growth performance parameters including weight gain (WG), specific growth rate (SGR), condition factor (CF), feed intake (FI), feed efficiency ratio (FER), and survival rate (SR) were calculated according to the following formulae: WG(%) = W_2_ − W_1_; SGR = 100(lnW_2_ − lnW_1_)/T, where W_1_ is the initial weight of the fish, W_2_ is the final weight of the fish, and T is the time duration; CF = weight of the fish (g)/(length of the fish)^3^(cm)^3^ × 100; FI (g/fish/7 months) = (dry diet given-dry remaining diet recovered)/number of fish; and FER = WG/FI, SR = (N_f_/N_0_) × 10, where N_0_ is the initial number of fish and N_f_ is the final number of fish.

Biochemical quality indices of moisture concentration (AOAC Official Method 930.15), the crude concentration of protein (AOAC-968.08), free fatty acids, and ash (AOAC-942.05) for commercial feed and iced fish were assessed, and the analyses were performed using standard methods. Briefly, muscle tissues were dried to a constant weight at 105 °C to determine their moisture content, while the crude protein content was determined using the Kjeldahl method (using N to protein conversion factor of 6.25). The total lipid content was extracted using the chloroform-methanol method. We determined the fatty acid composition of muscle tissue as described previously [1,10], based on each sample and three replicates for each group. Fatty acid content was expressed as mg/100 g (Appendix A).

Lipids from fresh muscle tissues were extracted by adding chloroform, dispersed via ultrasonication, and transferred to a gas chromatography-mass spectrometry (GC-MS) sample bottle (Agilent, Wentzville, MO, USA). Thirty-seven fatty acid standards (batch number: XA19945V) were purchased from Supelco (St. Louis, MO, USA). Chromatographically pure hexane and methanol were purchased from Merck (St. Louis, MO, USA), and 14% boron trifluoride-methanol was purchased from Macklin. The pretreatment standard for muscle sample comprised a 4 g live sample that was placed in a stoppered flask, followed by the sequential addition of pyrogallic acid, ethanol, and superoxide (Sigma-Aldrich, USA) with vigorous mixing. Concentrated hydrochloric acid was added and incubated in a 70 °C water bath for 40 min. A 1:1 solution of ether and petroleum ether was used for extraction and vacuum rotary drying (Buchi, Flawil, Switzerland) at 40 °C. A solution containing sodium hydroxide and methanol was added to condense the material, which was refluxed for 30 min. A solution of boron trifluoride and methanol solution was added with continued refluxing. In turn, hexane and saturated salt water were added. The sample was shaken and then kept stationary to allow particles to settle. The supernatant was collected and stored at −20 °C. Because of the high detection limit for fish fatty acid of common GC using an FID detector (approximately 1 mg/100 g), a model 7520 GC-MS, FID detector (Agilent, USA), and model CD-2560 chromatographic column (100 m × 0.25 mm × 0.20 μm) were used for detection. GC-MS analysis was performed using established standards and as described in the literature. The qualitative and quantitative analyses of samples were performed using the external standard method.

### 2.4. Sequencing and Taxonomy Classification

DNA extraction (MO BIO Laboratories for water and soil) and a two-step PCR was conducted to amplify the 16S rRNA gene [11] and ITS [12] for pyrosequencing. Simply, the V3-V4 region of the bacteria 16S ribosomal RNA gene was amplified via PCR (95 °C for 5 min, followed by 15 cycles at 95 °C for 1 min, 50 °C for 1 min, 72 °C for 1 min, and a final extension at 72 °C for 7 min) using primers (forward primer, 5′-ACTCCTACGGGAGGCAGCA-3′; reverse primer, 5′-GGACTACHVGGGTWTCTAAT-3′). The ITS primers were 5′-GCATCGATGAAGAACGCAGC-3′and 5′-TCCTCCGCTTATTGATATGC-3′. Total bacterial and fungous DNA were extracted from commercial feed (i.e., named as F and Fi for 16S rRNA and ITS, respectively) and iced fish; water (mixed) and intestinal fecal samples used both 16S rRNA and ITS sequence [10]. PCR amplification was performed in triplicate for each sample. All PCR products were quantified using the Quant-iT™ dsDNA HS Reagent and pooled. High-throughput sequencing analysis of bacterial rRNA genes was performed on the purified, pooled sample using the Biomarker Illumina NovaSeqplatform (Biomarker Technologies, Beijing, China, 2 × 250 paired ends) at Biomarker Technologies Corporation (Beijing, China). The operational taxonomic unit (OTU) analysis, abundance and diversity indices (Chao1, Ace, Shannon, and Simpson), and β-diversity analysis were performed following our previous studies [10]. The raw sequences were paired, spliced (FLASH, version 1.2.11), then quality filtered (Trimmomatic, version 0.33, and UCHIME, version 8.1) to obtain high-quality tag sequences. The high-quality tag sequences were clustered at >97% similarity (USEARCH, version 10.0), while the operating taxonomical units (OTUs) were filtered with 0.005% as the threshold. Silva (release 138) was selected for the bacterial 16S database, and Unite (release 8.0) was chosen for the fungal ITS database. Using the RDP Classifier for taxonomic annotation, the confidence threshold was 0.8 (version 2.2). The data for bacteria were analyzed using PyNAST (version 1.2.2), the fungus data were analyzed using ClustalW2, and the phylogenetic tree was created using the neighbor-joining method. The research was aimed at identifying the core microbiota in the same group and the presence of core bacteria and fungi in all the samples.

### 2.5. Change in Microbial Communities among Different Feeding Strategies

To clarify whether the compositions of the bacterial and fungal communities were correlated with or independent of different feeding strategies, OTU count data were used to construct dissimilarity matrices with the UniFrac. Significant increases or decreases in bacterial community were selected based on *p*-value from the sets of comparisons, e.g., PF (pond culturing using whole-course commercial feed) vs. WF (water tank culturing using whole-course commercial feed), PFi (pond culturing using commercial feed plus iced fish) vs. PIi (pond culturing using iced fish). The difference could differ at various levels, ranging from the phylum level to the species level, with the threshold of 0.5% [10].

To identify the microbial taxa branches, an analytical evolutionary branching graph (including a distribution histogram based on the LDA value) for cladogram branches among different treatment groups was drawn using LEFSe analysis, and then the network of the significant bacterial community was drawn using the SparCC algorithm, which is used for correlation analysis (including positive and negative correlations) and statistical testing. An analytical evolutionary branching graph for cladogram branches among the different treatment groups was drawn from July to September using LEFSe analysis. At the generic level, the G-TEST (for large samples, in which the number of functional genes annotated is more than 20) and Fisher test (for small samples, in which the number of functional genes annotated is less than 20) were used to test the significant difference between the two samples. Two t-tests were performed between different groups, and the COG/KEGG pathway was drawn. Python was (matplotlib) used for drawing the network diagram of species correlation, BugBase was used for predicting bio-level coverage of functional pathways within a complex microbiota, and bio-explanatory phenotypes such as oxygen tolerance, Gram-staining, and pathogenic potential were applied. In order to find the potential pathogenic bacteria/fungi source, the significant OTUs between feed, water (including front, middle, and back), and intestinal content were analyzed.

### 2.6. Data Analyses

The data of OTUs, fish growth parameter (WG, SGR, CF, FI, FER, SR), and concentrations of body composition were expressed as mean ± standard deviation (SD). Normal distribution of data was tested with Shapiro–Wilk test (α = 0.05). One-way analysis of variance (ANOVA) and Duncan’s post hoc test for normal data, as well as Kruskal–Wallis analysis for non-normal data, was used to test for differences among treatments (i.e., PF vs. PFi, PFi vs. PIi, etc.). Statistical tests were carried out using SPSS v. 26.0 (IBM Corp., Armonk, NY, USA).

## 3. Results

### 3.1. Water Quality and Disease Incidence

The DO, product yield, and other water quality indexes (SD, TSS, TN, TP) increased significantly (Appendix A, one-way ANOVA, F = 1.45, d.f. = 2.78, *p* = 0.014), and disease-related parameters (fatty liver disease rate and fat ratio, death rate except for the comparison of PF vs. WF) decreased significantly (Appendix A, one-way ANOVA, F = 3.42, d.f. = 2.66, *p* = 0.027) in WF compared with other groups. The ratios of disease time, death, fatty liver disease, and fat contents in PF were significantly lower than those in PI (one-way ANOVA, F = 1.65, d.f. = 2.23, *p* = 0.032, Appendix A). The DO in PFI were significantly higher than those in WF, while SD, TSS, TN, TP, and the ratios of death rate, fatty liver disease, and fat contents were lower (Appendix A). The DO and product yield in PF were significantly higher than those in WF, while SD, TSS, TN, TP, fatty liver disease, and fat contents were lower (Appendix A). Compared with the iced fresh fish group, the fluctuation of ammonia nitrogen in the pellet feed group was relatively small and the value was slightly lower, as was the total phosphorus. The frequency and duration of cyanobacteria outbreaks in the pellet feed group were significantly lower than those in the iced fresh fish group (Appendix A).

### 3.2. Growth Performance

There were no significant differences in moisture content (30.18–48.46%), crude protein (23.18–46.85%), crude lipid (4.73–5.92%), crude ash (1.47–2.73%), crude fiber (0.54–3.81%), together with WG, SGR, CF, FI, FER, and SR observed among all groups (Appendix A, *p* > 0.05). Crude protein and ash in iced fish were significantly decreased (*p* < 0.05) compared with commercial feed (Appendix A, one-way ANOVA, F = 3.48, d.f. = 1.77, *p* = 0.008). The product yields in PFI were significantly higher than those in WF (Appendix A, one-way ANOVA, F = 1.66, d.f. = 2.44, *p* = 0.009).

### 3.3. Body Composition

The unsaturated fatty acids (UFAs) consisted of three monounsaturated UFAs (MUFAs) and eight polyunsaturated UFAs (PUFAs, including seven highly UFAs named as HUFAs: four n-3 and three n-6 PUFAs). ∑SFA, ∑MUFA, ∑n-6PUFA, and n-6/n-3 in PF (commercial feed) were significantly higher than those in PI (iced fish with the lowest values, Figure 2, *p* < 0.05), while ∑n-3PUFA and ∑HUFA were significantly lower. The three main SFAs (C14:0 + C16:0 + C18:0) displayed higher values in PF, and methyl oleate (C18:1n9c) determined the larger values of MUFAs in PF than those in other groups (Figure 2, one-way ANOVA, F = 2.47, d.f. = 3.66, *p* = 0.064). Cis-4,7,10,13,16,19-docosahexaenoic acid methyl ester (C22:6n3) and methyl linoleate (C18:2n6c) were found to be the main constituents of ∑n-3PUFA and ∑n-6PUFA when a comparison was made among the different feeding strategy groups (Figure 2, one-way ANOVA, F = 2.33, d.f. = 2.28, *p* = 0.053). However, elaidic acid methyl ester (C18:1n9t), erucic acid (C22:1n9), linolenic acid methyl ester (C18:3n3), eicosatrienoic acid methyl ester (C20:3n3), and linolenic acid (C18:3n6) were the main constituents of ∑MUFA, ∑n-3PUFA, and ∑n-6PUFA (Figure 2, one-way ANOVA, F = 2.87, d.f. = 3.01, *p* = 0.072).

The unsaturated fatty acids (UFAs) consisted of three monounsaturated UFAs (MUFAs) and eight polyunsaturated UFAs (PUFAs, including seven highly UFAs named as HUFAs: four n-3 and three n-6 PUFAs). ∑SFA, ∑MUFA, ∑n-6PUFA, and n-6/n-3 in PF (commercial feed) were significantly higher than those in PI (iced fish with the lowest values, Figure 1B, *p* < 0.05), while ∑n-3PUFA and ∑HUFA were significantly lower. The three main SFAs (C14:0 + C16:0 + C18:0) displayed higher values in PF, and methyl oleate (C18:1n9c) determined the larger values of MUFAs in PF than those in other groups (Figure 2, one-way ANOVA, F = 2.47, d.f. = 3.66, *p* = 0.064). Cis-4,7,10,13,16,19-docosahexaenoic acid methyl ester (C22:6n3) and methyl linoleate (C18:2n6c) were found to be the main constituents of ∑n-3PUFA and ∑n-6PUFA when a comparison was made among the different feeding strategy groups (Figure 2, one-way ANOVA, F = 2.33, d.f. = 2.28, *p* = 0.053). However, elaidic acid methyl ester (C18:1n9t), erucic acid (C22:1n9), linolenic acid methyl ester (C18:3n3), eicosatrienoic acid methyl ester (C20:3n3), and linolenic acid (C18:3n6) were the main constituents of ∑MUFA, ∑n-3PUFA, and ∑n-6PUFA (Figure 2, one-way ANOVA, F = 2.87, d.f. = 3.01, *p* = 0.072).

Concerning the same feed, ∑SFA, ∑MUFA, ∑n-3PUFA, and ∑HUFA in PF (pond) were significantly higher than those in WF (water tank with the lowest values, Figure 2, *p* < 0.05), while ∑n-6PUFA and n-6/n-3 were significantly lower. When we compared groups with different culturing models, ∑SFA, ∑n-3PUFA, and ∑HUFA in PFI (feed plus iced fish) were significantly higher than those in WF, while ∑n-6PUFA and n-6/n-3 were lower. ∑SFA and ∑MUFA levels in PFI were somewhere between the values of ∑SFA and ∑MUFA observed in PF and PI. In total, ∑n-3PUFA revealed ordering as PI > PFI > PF > WF, with ∑n-6PUFA and n-6/n-3 displaying the reverse order of WF > PF > PFI > PI, accompanying the reverse high ∑n-3PUFA and ∑n-6PUFA in fish fed with commercial feed and iced fish, respectively. ∑HUFA in PI and PFI was significantly higher than that in WF and PF, with the same increasing tendency being evident in the iced fish groups compared with the commercial feed groups (one-way ANOVA, F = 1.39, d.f. = 2.74, *p* = 0.014), even though no significant differences in ∑PUFA were observed among all groups (one-way ANOVA, F = 1.13, d.f. = 1.77, *p* = 0.02).

### 3.4. α- and β-Diversity

A total of 629 bacterial OTUs were identified in the different groups, while lower values were revealed in WF and PI than that in PF. Culturing density explained the largest fraction (38.55% along PC1 for 16S rRNA; 39.54% along PC1 for ITS) of the variation in β-diversity, while 23.00% along PC2 for 16S rRNA (12.11% along PC2 for ITS) accounted for feeding strategies, intestinal absorptivity rate of commercial feed and iced fish, the efficiency of feed, and other causes. With respect to the water and feed samples, the largest fraction along PC1 for 16S rRNA changed to 44.78% (45.32% for ITS), while that along PC2 for 16S rRNA changed to 31.70% (28.91% for ITS).

### 3.5. OTU Biomarkers and Their Related Analysis for the Associated Pathway

Fusobacteria (PF:WF:PI = 50.1:83.0:15.5), Proteobacteria (PF:WF:PI:PFI = 16.0:13.5:35.0:33.8), and Firmicutes (PF:WF:PFI = 25.8:3.5:56.0) were the most dominant phyla present in all samples. With the increased feeding of iced fish, the abundance of Firmicutes (PFI:PI = 56.0:3.5) was decreased significantly and Tenericutes increased significantly (PFI:PI = 5.5:40.7, Table 1, both *p* < 0.001).

(1)
*OTUs between pond commercial feed (PF) and iced fish feed (PI) groups*


In more detail, in the comparison of PF vs. PI, phylum Tenericutes; class Mollicutes; order Bacteroidales, Mycoplasmatales, Vibrionales; family Mycoplasmataceae and Porphyromonadaceae significantly increased in PI compared with that in groups fed with commercial feed (Figure 3A). The relative abundance of certain classes (Clostridia and Mollicutes), orders (Clostridiales and Mycoplasmatales), and families (Clostridiaceae and Mycoplasmataceae) was significantly decreased in PI relative to that in PF (one-way ANOVA, F = 1.22, d.f. = 3.17, *p* = 0.013). Chytridiomycota and Cryptomycota of water samples (high in PFWi, Figure 3C) significantly increased in PIWi; Ascomycota significantly increased in Fi; while Basidiomycota, Glomeromycota, Mortierellomycota, and Mucoromycota significantly increased in Ii. With respect to the feed, the enriched KEGG pathways including coenzyme transport and metabolism, energy production and conversion, and defense mechanisms increased in group I (Figure 4A), while the pathways of transport and mechanism (carbohydrate, lipid, inorganic ion, nucleotide, and amino acid) decreased in group I compared with those in group F. The enriched COG pathways included endocrine and metabolic diseases, the digestive system, and neurodegenerative and infectious diseases: bacteria decreased in PI, while carbohydrate metabolism was enriched in PF (Figure 4C). Interestingly, feeding with iced fish enhanced the KEGG pathways of energy metabolism and their related better follow-up effect, including amino acid and vitamin usage and the nervous and endocrine systems.

(2)
*OTUs for using different culturing densities between pond (PF) and water tank commercial feed (WF) groups*


For PF vs. WF with different densities, inter-individual differences with close similarity were shown in the treatment groups (except Firmicutes and Proteobacteria). Ten phyla were expressed in PF, and a significant increase in Fusobacteria was revealed in WF (Figure 3A, Table 1, one-way ANOVA, F = 3.11, d.f. = 2.10, *p* = 0.036). Further, a significant increase in Proteobacteria and Firmicutes was observed with the increasing amount of iced fish. The order Fusobacteriales, family Fusobacteriaceae, and genera *Enterobacter*, *Escherichia*, *Shigella*, *Klebsiella*, and *Sarcina* were characteristic branches of taxa in WF. *Cetobacterium*, *Plesiomonas*, *Clostridium sensu* stricto, and *Mycoplasma* showed a good relationship with each other. The infectious diseases (:Bacterial and :Parasitic) and metabolism of other amino acid COG pathways were enriched with higher culturing densities (WF vs. PF, Figure 4C). Remarkably, fish culturing with high density enhanced the KEGG pathways of energy metabolism, defense mechanism, and environmental adaption (drug and xenobiotics removal), but exhibited decreased pathways of nuclear metabolism and replication/repair compared with pond culturing.

(3)
*OTUs by using different feeding strategies between pond commercial and iced fish combined feed (PFI), and pond commercial (PF) and iced fish (PI) feed, water tank (WF) groups*


Bacteroidota and Verrucomicrobiota significantly increased, while Armatimonadota, Chlorofiexi, and Actinobacteria significantly decreased in PFI when compared with those in PF. With respect to water samples, Bacteroidota and Actinobacteria showed the same tendency with increases and decreases, respectively, while Verrucomicrobiota significantly decreased in PFI when compared with those in PF. In particular, the genera *Candidatus* (*Methylopumilus* and *Aquirestis*) significantly increased in PFI, but the genera *Candidatus* (*Planktophila*) and *Limnohabitans* significantly decreased. With respect to the water, the enriched COG pathways including transport and mechanism (lipid, nucleotide, secondary metabolites, and amino acid), energy production and conversion, and defense mechanisms increased in the group combining feed and iced fish, while two KEGG pathways of transport and mechanism (carbohydrate, inorganic ion), reduced energy supply and reduced resistance to bacterial diseases.

Conversely, certain classes (Clostridia and Mollicutes), orders (Clostridiales and Mycoplasmatales), and families (Clostridiaceae and Mycoplasmataceae) were significantly increased in PFI relative to those in PI (Table 1, one-way ANOVA, H = 3.48, d.f. = 2.55, *p* = 0.007). For disease and transformation of bacteria and fungi, like the comparison in F vs. I, 13 genera *Bacillus* 25 *Rhodoferax* significantly increased in commercial feed compared with iced fish. The carbohydrate metabolism pathway was compared among PFI and PI.

Clostridiaceae, Mycoplasmataceae, and Fusobacteriaceae more significantly increased and later decreased in PFI than those in WF, respectively. The bacteria *Candidatus* more significantly increased and *Limnohabitans* more significantly decreased in PFIW (water in PFI) than those in WFW (water in WF, Figure 3B), respectively. The fungi *Stereum*, *Zygorhizidium*, and *Tremella* significantly increased in PFIW. The COG/KEGG pathways including biosynthesis of other secondary metabolites, nucleotide metabolism, environmental adaption, replication and repair, immune diseases, and endocrine and metabolism diseases increased in the group combining feed and iced fish, while for membrane transport, infectious diseases (:Bacterial and :Parasitic), and neurodegenerative diseases, the branch in the pathway of digestive system increased in the group with high density only using feed without iced fish (Figure 4B,C).

(4)
*The pathogenic bacteria and fungi between groups and their related characteristic branches*


Basidiomycota and Ascomycota in PFIW and PIW were significantly higher than those in PFW and WFW, while in genera *Stereum*, *Zygorhizidium*, and *Tremella* they showed the same tendency with a significant increase. The bacterial branch in PIW, especially Sporichthyaceae and Pelagibacteraceae, was significantly higher than that in PFIW, PFW, and WFW, while Comamonadaceae, Pirellulaceae, and Alcaligenaceae significantly decreased. The fungus in Ii was different from that in Fi, while the fungus in PIWi was different from other water groups, and Ganodermataceae increased in PIW. The genera *Stereum*, *Zygorhizidium*, and *Tremella* significantly increased in Ii compared with the other groups. In total, 22 genera *Aspergillus* and 8 *Ganoderma* significantly increased in Ii.

The *Rhabdoviruses*, *Iridovirus* and the bacteria *Nocardia*, *Edwardsiellasis*, *Aeromonas*, *Flavobacterium columniformis*, and *Aeromonas veronii* showed no significant difference between different water, feed, and fecal samples. The bacterial branch of *Cetobacterium*, *Plesiomonas*, *Clostridium*, *Mycoplasma*, fungi of *Ganoderma*, and *Aspergillus* showed the highest correlation within the significant enriched KEGG pathways. When we compared this with water and feed samples, *Candidatus* and *Stereum* revealed as the highest, and after adding fecal samples, *Bacillus* and *Rhodoferax* occurred as the highest correlation relationship (Figure 5). Based on the function analysis, mobile elements (with pathogenic spread), facultative anaerobic/aerobic (producing harmful secondary metabolites), biofilms and stress tolerant (nutrient removal), and Gram-negative and potential pathogenic bacteria increased when iced fish were used as feed (Figure 5D), while Gram-positive anaerobic bacteria decreased. With respect to the high culturing density, the bacteria became lower except for Gram-positive (recommendation of antibiotic use). When we used the mixed feeding strategy, we could slow down the amount of Gram-positive bacteria, but enhanced the abundance of Gram-negative potential pathogens, biofilms, prestress-tolerant bacteria, and the medium mobile elements (recommendation of precise chemical use in disease and nutrient control).

(5)
*The potential pathogenic bacteria/fungi source*


The bacteria/fungi OTUs for feed, intestine, water, front, middle, and back were 1216/663, 333/1602, 43,627/6896, 16,480/3053, 13,680/2381, and 14,816/2428, respectively. The different expressed characteristic branches for bacteria in the feed (Patescibacteria, Myxococcota, Deinococcota, Methylomirabilota, Acidobacteriota, and Nitrospirota), intestine (Fusobacteriota increased; while Bdellovibrionota, Actinobacteriota, Thermotogota, Gemmatimonadota, Dependentiae, Bacteroidota, Proteobacteria, and Cyanobacteria decreased), front (Spirochaetota, Cloacimonadota, Fermentibacterota, Desulfobacterota, and Campylobacterota), middle (Actinobacteriota and Thermotogota), and back (Latescibacterota, Hydrogenedentes, Armatimonadota, Chloroflexi, and Cyanobacteria) were isolated (Figure 6). *Limnohabitans* from front water related to the nitrogen cycle demonstrated the highest correlation relationship. The increased intestinal bacterial species of *Mycoplasma*, *Plesiomonas shigelloides*, and *Cetobacterium somerae* and decreased *Comamonadaceae* and *Candidatus Fonsibacter ubiquis* were displayed as the specific branch. For fungi, characteristic branches in intestine (Ascomycota, Mortierellomycota, Olpidiomycota, Zoopagomycota, Glomeromycota, Mucoromycota, Kickxellomycota, Basidiobolomycota, Neocallimastigomycota, and Entomophthoromycota), feed (Basidiomycota), front (Aphelidiomycota and Monoblepharomycota), middle, and back (Rozellomycota) are presented. An increase in fungous species of Ascomycota and Basidiomycota and a decrease in Chytridiomycota was revealed in feed and intestine. Bacteria Cetobacterium/Clostridium sensu stricto/Plesiomonas in intestine, Paucibacter in feed, and fungi *Stereum* in feed were the specific branch, and most pathogenic bacteria presented in the middle and back area of the water, while intestinal pathogenic fungi revealed a similar expression with that in the front area of the water. The closer similarity between samples of feed and intestine, whole water, and front area of the water are shown via UPGMA clustering tree analysis (Appendix A).

## 4. Discussion

### 4.1. Changes for the Components of Fatty Acid

Fish cannot synthesize n-3 and n-6 HUFA from scratch because of the lack of Δ12 and Δ15 desaturases [7]. C18PUFA promotes the transcription of key enzymes in HUFA synthesis, while some types of n-3 HUFA inhibit transcription. There is a metabolic competition between n-3 PUFA and n-6 PUFA [13]. This is because n-3 PUFA and n-6 PUFA are both substrates of Δ6 desaturase during fish fatty acid metabolism, and there is competitive inhibition between these substrates [11]. Because the linolenic acid is oxidized quickly in vivo, it is not preserved like linoleic acid (18:2n-6), and the n-3 long-chain PUFA (LC-PUFA) produced by linoleic acid integrates into the cell membrane faster than the n-6 LC-PUFA produced by linoleic acid. A treatment with a high proportion of 18:3n-3/18:2n-6 can increase the activity of desaturase and enhanced 22:6n-3 with soya-bean oil administration [14]. Higher ∑SFA, ∑MUFA, ∑n-3PUFA, and ∑HUFA were evident upon feeding with iced fish compared with the commercial feed (with higher ∑n-6PUFA). After feeding, the fatty acid composition of LB not fed iced fish exhibited higher ∑SFA, ∑MUFA, ∑n-6PUFA, and 18:3n-3/18:2n-6, while ∑n-3PUFA and ∑HUFA were enriched in fish fed with commercial feed. Feeding with iced fish enhanced the KEGG pathways of energy metabolism, disease prevention, and its related better effect both in content and the related surrounding water. Combing the detail of the increased potential pathogenic bacteria (Figure 4) with iced fish administration, all these findings suggest a higher carbohydrate and fatty acid metabolism efficiency in the LB fed iced fish. It is undeniable that LB fed with fresh fish are more suitable for human consumption to obtain more polyunsaturated fatty acids, which may offer an additional diet strategy for obtaining more LC-PUFAs.

### 4.2. Different Feeding Strategies Shaping Body Composition, Gut Microbiota, and Enhancing Pathogenic Prevention

The intestines of aquatic animals play an important role in a variety of physiological functions such as the digestion and absorption of nutrients and immune activities. However, the intestinal microbiota is susceptible to changes in the external environment. Environmental factors such as water quality (DO, some toxicants such as heavy metals) and treatment composition (such as zooplankton enhanced *Cetobacterium* and *Rhizobium*, phytoplankton, etc.) [5,10,15], nutritional enzyme activity, and disease may shape gut microbiota [16]. A multi-omics strategy may help to understand the mode of action which is performed in fish species [2,3,9,10,12,15,16]. The crosstalk between gut microbiota, growth performance, quality of aquatic product, and even disease prevention may help us deeply understand the difference between feeding strategies performed by fish farmers (Figure 7).

(1)
*OTUs related to nutrient absorption efficiency between pond commercial feed (PF) and iced fish feed (PI) groups*


When fish were exposed to aluminum, the Porphyromonadaceae (higher in PI) significantly increased, while it decreased after probiotic supplementation [17]. Species belonging to Porphyromonadaceae families linked to volatile fatty acid (number of C < 10) consumption increased in their presence during the recovery period (fewer ∑SFA in PI). Tenericutes significantly decreased (*Coreius guichenoti*) [18], but increased in PI compared with PF in the present study. The pathogeny in iced fish and commercial feed was determined, and the bacterial branch of the water samples from iced fish was the same as those in the gut, except for Alcaligenaceae. A previous study showed Alcaligenaceae (increased in fresh fish) [19] decreased in air- and vacuum-packed crisp grass carp, while Comamonadaceae increased. The decreased gut Alcaligenaceae and *Limnohabitans* were found to be connected with NH_3_-N oxidation [20] and found in feed after adding pomegranate peel for preventing disease [21], which hinted Alcaligenaceae as pathogenic bacteria, and can be reduced with clean water or by nutritional immunity enhancer administration. That means gut Alcaligenaceae came from eating the iced fish, and it suggests that Alcaligenaceae had a relationship with disease in LB. The input (front) and drain-off (back) water affected the component of fish gut microbiota, especially the pathogenic bacterium; even the functional bacterium associated with nutrient substance absorption can be affected by the feed and the water lived in the past [1,9]. Results showed that fish intestinal *Mycoplasma*, *Plesiomonas shigelloides* (pathogen in grass carp) [22], and *Cetobacterium somerae* increased with decreased *Comamonadaceae*, and *Mycoplasma* may result in the impairment of intestinal health in LB [23], neoplasms or preneoplastic lesions [24], and mucosa-associated metabolic regulator [25].

There was no significant difference in water quality between PF and PI, but PI had higher annual morbidity, mortality, fatty liver incidence, and fat ratio than PF. Mycoplasmataceae and Comamonadaceae were found to be associated with carotenoid synthesizing, which determined the fillet color [26], while Porphyromonadaceae were involved in “alpha-linolenic acid metabolism” and “fatty acid biosynthesis” pathways. A recent study showed that *Cetobacterium somerae*, an acetate producer, was related to glucose homeostasis [27]. The difference in feed related to the digestive system [28] may contribute to the significant microbiota branch in the fish gut, which, verified in the present study, showed significant enriched KEGG pathways in PI (Figure 4C).

(2)
*OTUs related to environmental adaption by different culturing density between pond (PF) and water tank commercial feed (WF) groups*


For well water quality (higher DO with lower SD/TSS/TN/TP) and higher product yield in WF, a consistent association was observed between β-diversity of the gut microbiota and dissolved oxygen concentration in water when compared with PF [29]. However, except for the better water quality, the culturing density changed the structure of the gut microbiota. Fusobacteria increased in WF related to the enriched pathways of energy metabolism, defense mechanism, and environmental adaption. The pathogeny in commercial feed/iced fish and intestine was determined, and the results showed that *Stereum* was specific in feed. The present study shows that the input and outflow of water affect the intestinal flora in the surrounding water and in the gut, which in turn affects growth and disease resistance.

(3)
*The pathogenic bacteria and fungi OTUs by different feeding strategies between pond commercial and iced fish combined feed (PFI) and other groups*


The replacement of fish oil with vegetable oil reduced the formation of primary oxidation products, but compounds in the marine ingredients might affect protein oxidation [26]. Feeding regimes [30] (*Acipenser baeri*) significantly influenced fatty acid composition, but the differences in amino acid composition were significant (∑n-3PUFA increased, but ∑n-6PUFA decreased) in the present study. Pond field experiments were verified with confirmation from experienced fishermen stating that LB fed with iced fresh fish tasted good, which hinted that the synthesis signal pathway of aromatic odor ingredient increased based on our muscular composition and KEGG pathway analysis (biosynthesis of other secondary metabolites boosted). The chitosan (metabolic panel point) and probiotic-fed fish displayed a significantly higher abundance of Firmicutes, Bacillus, Enterococcus, and Pediococcus [16], corresponding to enhanced fish gut immunity (beneficial microbial populations, goblet cell density, villi length, and transcriptional regulation). Firmicutes and Tenericutes [31] were highly significantly decreased and increased, and pathways of carbohydrate metabolism, digestive system, and infectious disease: Bacterial (iced fish prevent disease) [32] were enriched in the groups fed with the commercial feed and iced fish. The well water quality (higher DO with lower SD/TSS/TN/TP) and higher product yield resulted in a lower death rate/fatty liver disease rate/fat ratio in WF when compared with that in PFI.

Gut structure (long or short), gastrointestinal shaping, and feed derived from water and/or treatment may play a role in shaping the fish intestinal microbiota [9,31]. Excessive dietary starch in feed can result in oxidative stress and suppress innate immunity and health [32]. The identification of potential probiotic/pernicious bacteria in the intestinal microflora at different growth rates demonstrated that Flavobacterium and Corynebacterium were the dominant bacteria in water and feed, respectively [33]. *Cetobacterium*, *Plesiomonas*, and *Clostridium* were the predominant genera in the intestine of fast- and medium-growing *Anguilla marmorata* [34]. Rhodoferax and Mycoplasmataceae were significantly increased in PFI relative to those in PI, and the abundance of Bacteroidales, Mycoplasmatales, and Vibrionales was significantly increased in PI compared with fish receiving commercial feed with disease [35].

During the first feeding stage, Rhodoferax increased [36] and was essential to the removal of nitrogen [37]. Clostridia was significantly increased in PFI (vs. PI), similar to infected common carp [38]. The pathways of metabolism of other amino acids were enriched WF (higher density causing disease and fatty liver) when compared with PF, indicating that the order of energy sources (carbohydrate, fatty acid, protein) is affected with higher density. This suggested the immunity may be improved by improving water quality when a treatment of frozen fish is replaced by expanded feed [39] or using the combined commercial feed and iced fish, based on pursuing rapid growth under higher culturing density. The ability of probiotics to modulate fish humoral/mucosal immunity and host microbiota [10] has been studied, even after compromised situations such as microbial infections or stress [38]. Probiotics may boost the quality of both the fish (e.g., growth, well-being, and health status) and the fish environment. *Bacillus* and *Aspergillus* [40] could modulate the gut immune response, fat metabolism bacterial assembly [41], and prevention. Nowadays, the structure of intestinal bacteria, pathogenic bacteria, and related nutrient absorption are affected through fecal microbial transplantation. The present study showed resveratrol administration shaped tilapia intestinal structure, flora [10], and immune enhancement [10,33,42] to prevent fatty liver. This study showed that the replacement of iced fish with feed did not affect amino acid composition, but altered amino acid metabolism and the relative abundance of intestinal microbiota in crab [43]. Whether the involved predicted pathways that this study obtained, what was the actual bacterial/fungous affected pathway? Our methods rely on reference genomes in databases that may not always match what is in the sampled ecosystem. Even metagenomic data identifying the presence/absence of bacterial genes do not guarantee that we know functionality in in vivo conditions. Multi-omics is definitely better, but not everyone has this capability. The change in feeding strategy to reduce the disease incidence and fatty liver rate of LB and how to use substances from Chinese medical herbs or prebiotics is worthy of further study.

## 5. Conclusions

It is unclear how different feeding strategies or habitats may differentially shape the gut microbiome over the short- and long-term and then affect muscle growth quality and body composition. This study investigated the effects of dietary commercial feed, iced fish, and a combination of both on different metabolic parameters of the largemouth bass. Results showed no significant differences found in the growth performance except for the product yield using a different culture mode (PFI vs. WF). The fatty acid composition in LB fed a treatment without iced fish was richer in ∑SFA, ∑MUFA, ∑n-6PUFA, and 18:3n-3/18:2n-6. ∑n-3PUFA and ∑HUFA were enriched in LB fed with commercial feed, while for the gut microbiota, Firmicutes and Tenericutes were highly significantly decreased and increased, respectively, with the increasing amount of iced fish in the fish feed. Classes Clostridia and Mollicutes were significantly increased in the PFI group relative to those in the PI group. The KEGG pathways of carbohydrate metabolism, digestive system, and infectious diseases: Bacteria were enriched in the groups fed using commercial feed and iced fish. However, the higher rate of death, fatty liver disease, and frequency and duration of cyanobacteria outbreaks increased in PI. The present findings suggest that fatty acid metabolism efficiency is greater in LB fed iced fish, and that feeding with iced fish probably enhances the prevention of infectious diseases: Feeding with iced fish resulted in more activities in the digestive system and energy metabolism, more efficient fatty acid metabolism for owning the higher ∑MUFA, and simultaneously had potential protection for infectious bacteria from the environment through the change for intestinal microbiota in the pond of largemouth bass culturing. Finally, the difference in feed related to the digestive system may contribute to the significant microbiota branch in the fish gut, and the input and outflow of water affects the intestinal flora in the surrounding water and in the gut, which in turn affects growth and disease resistance. From the perspective of animal health farming, we may need to adjust feeding strategies and strengthen the control of water and drug intake/drainage to adapt to the iced fresh fish replacement action proposed by the Ministry of Agriculture and Rural Affairs. The data from multi-omics may not always match what is in the sampled ecosystem; metagenomics data need to be further studied to make our result close to the reality.

## Figures and Tables

**Figure 1 microorganisms-11-01356-f001:**
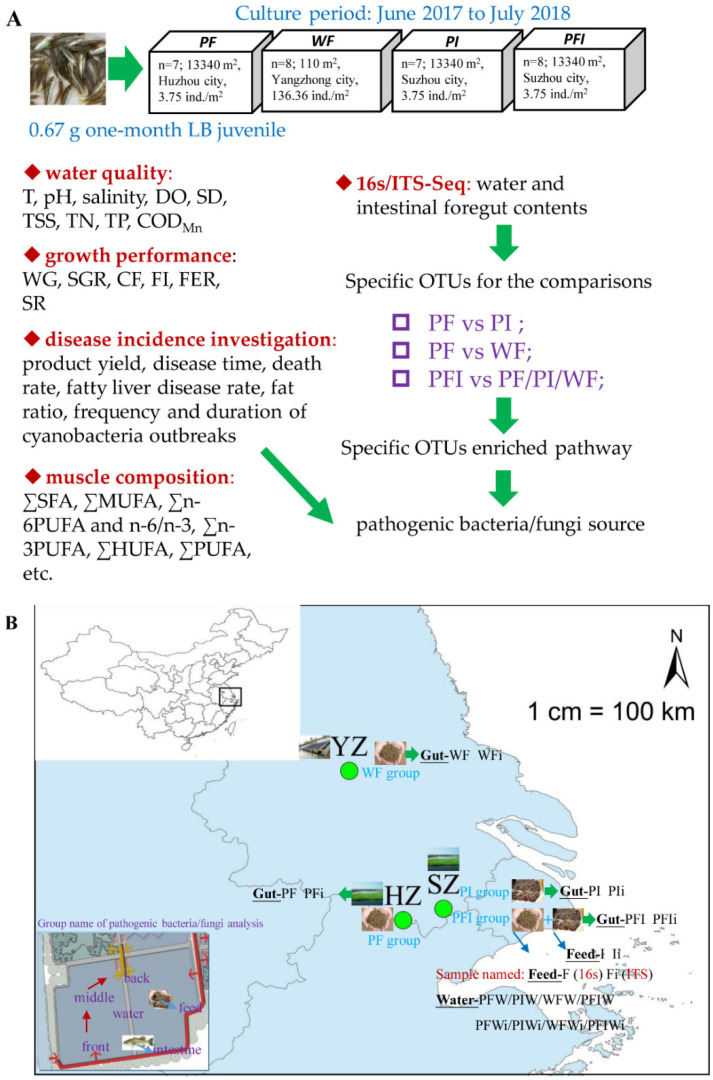
Scheme and sampling points. (**A**), the experimental design. (**B**), the three water tanks using whole-course commercial feed (5% feeding ratio) were designated as the WF group (water tank using feed; 110 m^2^, Yangzhong city, 32.297242° N, 119.80741° E). The three soil dike ponds using whole-course commercial feed were designated as the PF group (pond using feed; 13,340 m^2^, Huzhou city, 30.925088° N, 120.300707° E). The three ponds used for whole-course iced fish (Chuzhou Hengchuan Food Co., Ltd. (Chuzhou, China); 8% feeding ratio) were designated as the PI group (pond using iced fish; 13,340 m^2^, Pingwang of Suzhou city, 31.030636° E, 120.672257° N). Finally, the three ponds used for the commercial feed (8% feeding ratio for 5 months) and iced fish (8% feeding ratio for 6 months) were designated as the PFI group (13,340 m^2^, Pingwang). PF group (*n* = 7, sampled on 28 June 2018), WF group (*n* = 8, sampled on 26 June 2018), PI group (*n* = 7, sampled on 2 July 2018), and PFI group (*n* = 8, sampled on 2 July 2018). The primer set used to characterize the full dataset was selected because it was capable of enriching classifiable sequences while reducing the amplification of plastid-related sequences in the gut microbiota.

**Figure 2 microorganisms-11-01356-f002:**
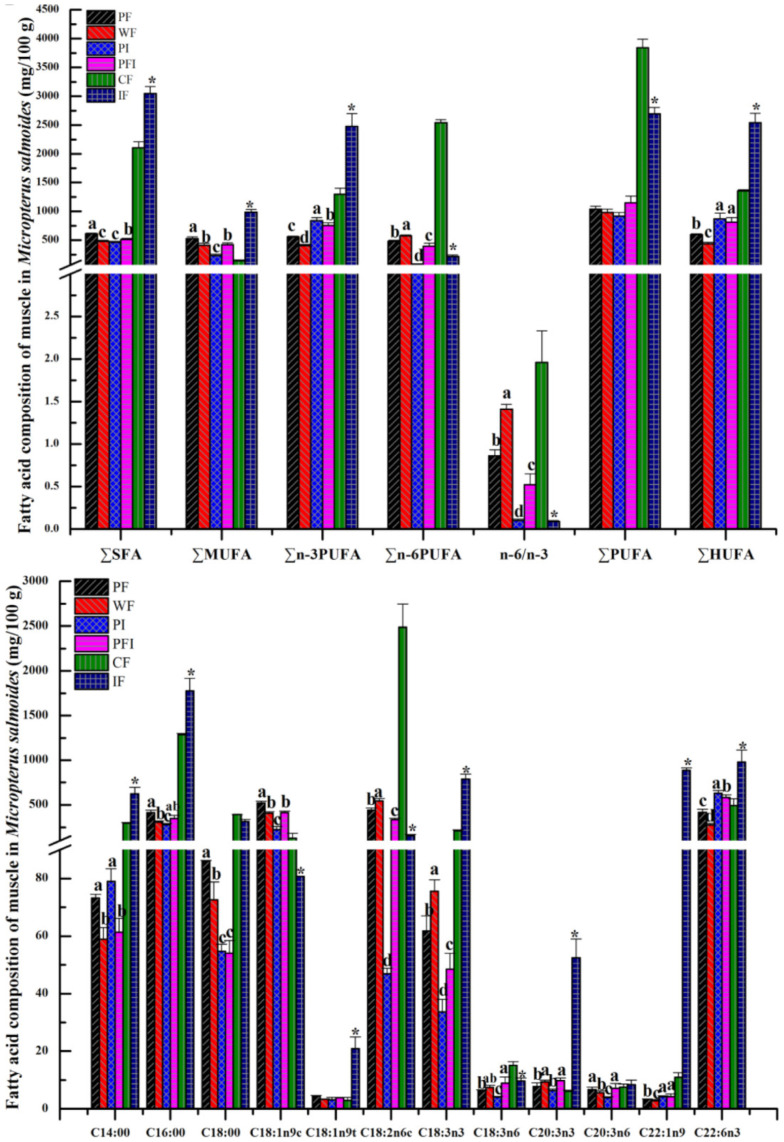
Fatty acid composition (mg/100 g) of muscle in *Micropterus salmoides* from different feeding strategy groups. CF and IF named as values for the commercial feed and iced fish. Values are presented as means ± SD using Origin 8.0 software. C4:0~C13:0, C15:0, C17:0, C20:0~C24:0, C20:5n3, C20:4n6 have not been revealed in the figure. * showed the significant difference between kinds of feeds, while the lowercase showed the significant difference among the groups.

**Figure 3 microorganisms-11-01356-f003:**
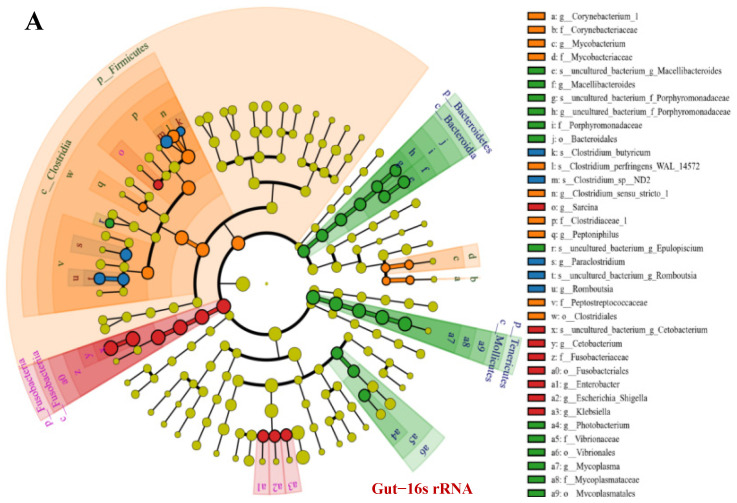
The significant characteristic branch histogram and different expressed OTUs with different feeding strategies. (**A**), Significant histogram among different treatment groups based on the LDA value (LDA score > 4.0). The length of the histogram represents the influence of different species based on the LDA score. Different colors represent the species of different groups. The circles radiating from the inside to the outside of the evolutionary branching map represent the classification level from phyla to species; each circle at different classification levels represents a classification at that level, and the diameter of the circles is proportional to the relative abundance. The yellow color is used to unify the species without significant differences, while the other species with a significant difference are colored according to the highest abundance for the groupings. Different colors represent different groupings, and nodes with different colors represent microorganisms that play an important role in the grouping. Taxonomic categories (**B**) are the results for 16S rRNA, while this is (**C**) for ITS. Taxonomic categories for each group at the phyla and species level and heat map of phylum abundance with clustering analysis are presented. The horizontal clustering is the sample information and the vertical clustering is species information; the left tree is the species clustering tree, the upper tree is the sample clustering tree, and the middle tree is the heat map.

**Figure 4 microorganisms-11-01356-f004:**
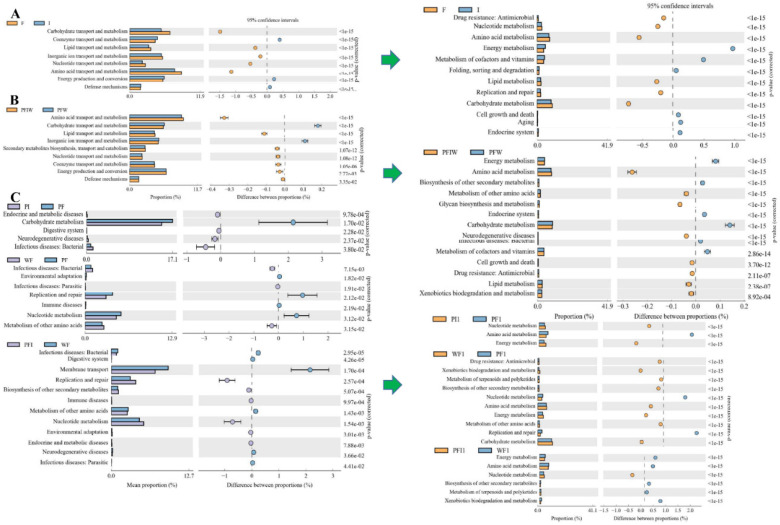
The classified COG and KEGG pathways between different samples. (**A**), feed samples for commercial feed and iced fish; (**B**), water samples in the comparison of PFIW and PFW; (**C**), fecal samples in the comparisons of PF vs. PI, PF vs. WF, PFI vs. WF. Different analyses of KEGG metabolic pathways reveal differences and changes of microbial community functional genes in the metabolic pathways between the different groups of samples. It is an effective means to study the functional metabolic changes of community samples that occur during acclimation to environmental changes. Differential analysis of KEGG metabolic pathways at the second level (also for the third and first level) is shown. Different colors represent different groupings. The left figure shows the abundance ratio of different functions in two samples or two groups of samples. The middle figure shows the difference ratio of functional abundances in 95% confidence interval. The rightmost value is the *p*-value.

**Figure 5 microorganisms-11-01356-f005:**
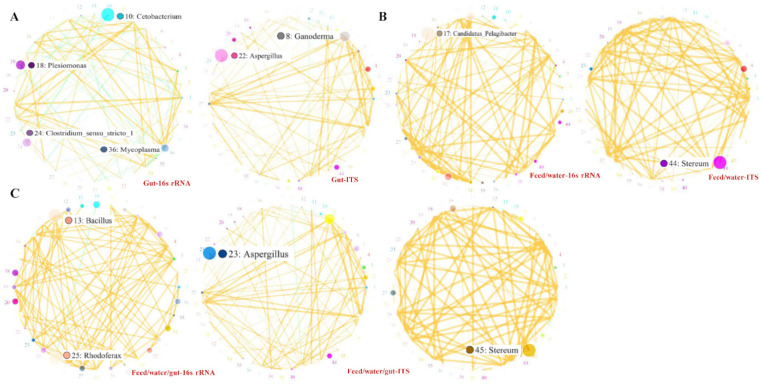
The highest correlation relationship within the significant enriched KEGG pathways. According to the abundance and variation of each species in each sample, the SparCC algorithm was used for correlation analysis (including positive correlation and negative correlation) and statistical testing. Datasets with correlation > 0.1 and *p* < 0.05 were screened, and co-expression was drawn based on the Python program. In the analysis of the network diagram, the characteristic branches are shown evenly through the existing top 50 genera with the highest correlation. (**A**–**C**) stand for content, feed/water, feed/water/content, respectively. (**D**), the characteristic branches within the significant enriched KEGG pathways via function analysis. The expression of the categorized mobile elements, facultative anaerobic, anaerobic, aerobic, biofilms, Gram-negative, Gram-positive, potential pathogenic bacteria, and stress tolerance is shown.

**Figure 6 microorganisms-11-01356-f006:**
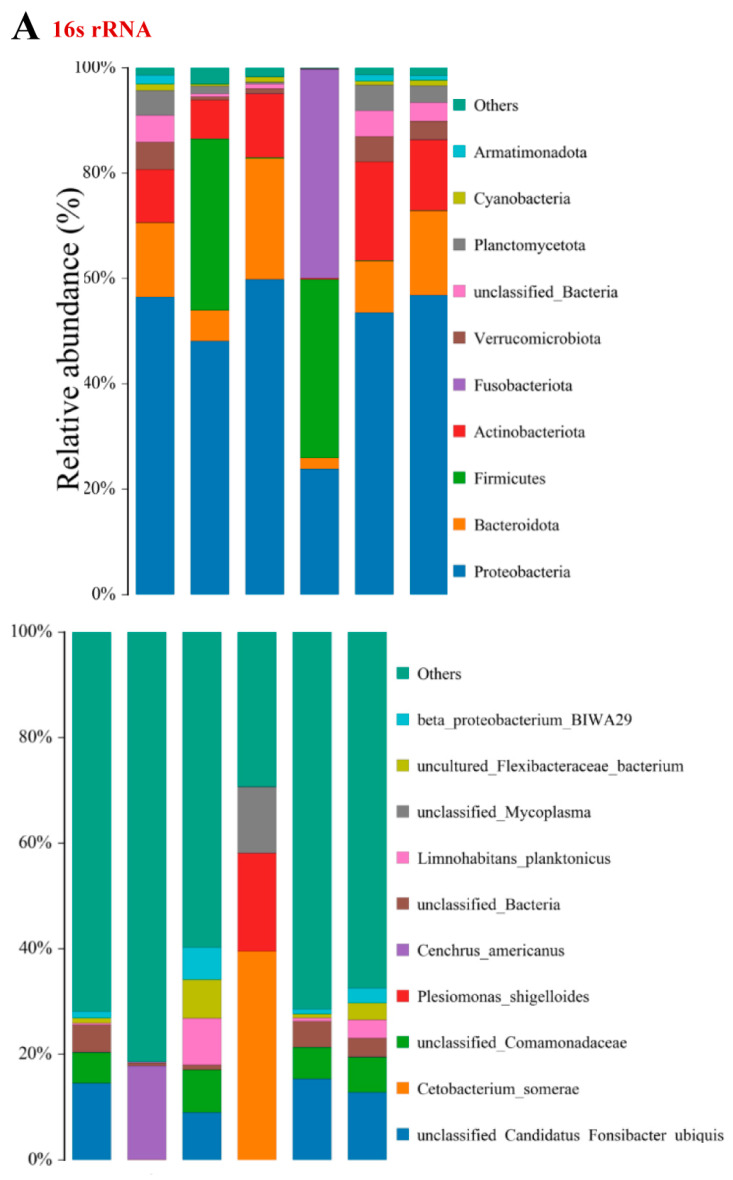
The different expressed characteristic branches of potential pathogenic bacteria/fungi source. Taxonomic category (**A**) is the result for 16S rRNA, while (**B**) is that for ITS. Distribution of species at phylum and species-level are shown in the following figures from left to right, respectively. Each color block represents a term in the corresponding taxonomic level. The area of each block represented relative abundance of the term. For an optimal appearance, top 10 species in each level were selected and the rest were combined as others. Unclassified represents the species that have not been taxonomically annotated. Detailed species information can be found in the species richness table of the corresponding classification. The similarity and difference of community composition among samples are reflected by color gradient. The heat maps were generated using the R package according to species composition and relative abundance at level of phylum classification. In heatmaps, the color represents species abundance. Clustering on the left side represents similarity of the phylum between the samples, i.e., phylum with closer distance or branch length share more similar abundance patterns across samples. Clustering on the top represents similarity of samples, i.e., samples with closer distance or branch length share more similar patterns of species.

**Figure 7 microorganisms-11-01356-f007:**
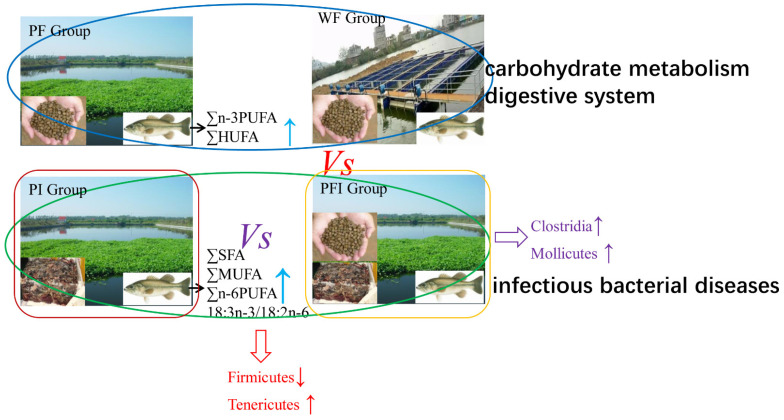
The overview of the present study.

**Table 1 microorganisms-11-01356-t001:** Statistically significant characteristic branches of different classifications among different treatment groups using Lefse analysis ^#^.

Classification	Name	PF (*n* = 7)	WF (*n* = 8)	PI (*n* = 7)	PFI (*n* = 8)	PI:PF (*p*)	WF:PF (*p*)	PFI:PF (*p*)	PFI:PI (*p*)	PFI:WF (*p*)
Phylum	Firmicutes	0.2577 ± 0.3333	0.0240 ± 0.0180	0.0345 ± 0.0497	0.5600 ± 0.3001				##	##
	Fusobacteria	0.5014 ± 0.2783	0.1553 ± 0.0908	0.8297 ± 0.2920	0.0373 ± 0.0820	*	*	**		**
	Tenericutes	0.0575 ± 0.0634	0.4069 ± 0.3253	0.0004 ± 0.0004	0.0553 ± 0.1481	**			**	**
Class	Clostridia	0.2481 ± 0.3294	0.0228 ± 0.0178	0.0117 ± 0.0113	0.5523 ± 0.3116				##	##
	Fusobacteria	0.5014 ± 0.2783	0.1553 ± 0.0908	0.8297 ± 0.2920	0.0373 ± 0.0820	*	*	**		*
	Mollicutes	0.0575 ± 0.0634	0.4069 ± 0.3253	0.0004 ± 0.0004	0.0553 ± 0.1481	**			**	**
Order	Clostridiales	0.2481 ± 0.3294	0.0228 ± 0.0178	0.0117 ± 0.0113	0.5523 ± 0.3116				##	##
	Fusobacteriales	0.5014 ± 0.2783	0.1553 ± 0.0908	0.8297 ± 0.2920	0.0373 ± 0.0820	*	*	**		*
	Mycoplasmatales	0.0575 ± 0.0634	0.4069 ± 0.3253	0.0004 ± 0.0004	0.0553 ± 0.1481	**			**	**
Family	Clostridiaceae_1	0.1679 ± 0.3511	0.0202 ± 0.0167	0.0027 ± 0.0034	0.4681 ± 0.3159				**	**
	Fusobacteriaceae	0.5014 ± 0.2783	0.1553 ± 0.0908	0.8297 ± 0.2920	0.0373 ± 0.0820	*	*	**	**	**
	Mycoplasmataceae	0.0575 ± 0.0634	0.4069 ± 0.3253	0.0004 ± 0.0004	0.0553 ± 0.1481	**			**	##
Genus	Cetobacterium	0.5014 ± 0.2783	0.1551 ± 0.0906	0.8297 ± 0.2919	0.0373 ± 0.0820	*	*	**		*
	Clostridium_sensu_stricto_1	0.1556 ± 0.3557	0.0182 ± 0.0160	0.0012 ± 0.0022	0.4659 ± 0.3193				**	**
	Mycoplasma	0.0575 ± 0.0634	0.4069 ± 0.3253	0.0004 ± 0.0004	0.0553 ± 0.1481	**			**	**
Species	Clostridium_perfringens_WAL-14572	0.0000 ± 0.0001	0.0013 ± 0.0028	0.0005 ± 0.0013	0.0807 ± 0.0661			##	##	##
	uncultured_bacterium_g_Cetobacterium	0.5014 ± 0.2783	0.1551 ± 0.0906	0.8297 ± 0.2919	0.0373 ± 0.0820	*	*	**		*

^#^ Statistically significant and highly significant effects are denoted by an asterisk (*, *p* < 0.05) and double asterisk (**, *p* < 0.01), respectively, and double ‘#’ indicates a *p*-value < 0.001.

## Data Availability

Not applicable.

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
