# Peer review of "The Effects of Different Feeding Regimes on Body Composition, Gut Microbial Population, and Susceptibility to Pathogenic Infection in Largemouth Bass"

_microorganisms, 2023, doi:10.3390/microorganisms11051356_

Round 1

Reviewer 1 Report

Microorganisms (microorganisms-2363910). “Different feeding strategies shape body composition, gut microbiota, and susceptibility to pathogenic infection in largemouth bass”

Reviewer comments:

In general, the present manuscript: “Different feeding strategies shape body composition, gut microbiota, and susceptibility to pathogenic infection in largemouth bass” by Zheng and colleagues focused to investigate the effects of dietary administration of commercial feed and/or iced fish on various parameters including growth performance, body composition, and gut microbiota, and it has an interesting goal. On the other hand, some specific comments are given bellow, to help improving the quality of the manuscript reviews.

Final remark: this current manuscript needs minor review.

General comments

- In general, I really appreciate this work with careful design and abundant data. In addition, it’s important to note that the manuscript is well written (English grammar and composition). Congratulations.

Specific comments

Abstract section:

- Conclusion must be made based on the premises described at the final of introduction section. Conclusion strictly related to the results obtained. Depending on the assumptions/suggestions/probability/hypothesis, it can be considered in the discussion. Review the entire manuscript.

- Usually, key words are words that do not contain in the title of the manuscript. Review the entire of manuscript.

Introduction section:

- Currently, most of the scientific manuscript are presented as hypotheses to be more attractive and interesting than description of goals. The present manuscript can be presented with hypothesis. We suggest the authors to present this manuscript with more attractive hypothesis and to make the manuscript more interesting.

Conclusion

(again): Conclusion must be made based on the premises described at the final of introduction section. Conclusion strictly related to the results obtained. Depending on the assumptions/suggestions/probability/hypothesis, it can be considered in the discussion. Review the entire manuscript.

Figures: poor quality!

Author Response

In general, the present manuscript: “Different feeding strategies shape body composition, gut microbiota, and susceptibility to pathogenic infection in largemouth bass” by Zheng and colleagues focused to investigate the effects of dietary administration of commercial feed and/or iced fish on various parameters including growth performance, body composition, and gut microbiota, and it has an interesting goal. On the other hand, some specific comments are given bellow, to help improving the quality of the manuscript reviews.

Final remark: this current manuscript needs minor review.

General comments

- In general, I really appreciate this work with careful design and abundant data. In addition, it’s important to note that the manuscript is well written (English grammar and composition). Congratulations.

Specific comments

Abstract section:

- Conclusion must be made based on the premises described at the final of introduction section. Conclusion strictly related to the results obtained. Depending on the assumptions/suggestions/probability/hypothesis, it can be considered in the discussion. Review the entire manuscript.

Response: Thank you very much for the comments. “Results showed that no significant differences found in the growth performance except for the product yield using different culture mode (PFI vs WF)…... For muscle composition,…While for the gut microbiota,…, corresponding with the higher rate of death, fatty liver disease, frequency and duration of cyanobacteria outbreaks…” has been added in the revised version in line 22-23, 24, 26, 33-34, etc.

- Usually, key words are words that do not contain in the title of the manuscript. Review the entire of manuscript.

Response: Thank you very much for the comments. The key words have been changed in line 42-3.

Introduction section:

- Currently, most of the scientific manuscript are presented as hypotheses to be more attractive and interesting than description of goals. The present manuscript can be presented with hypothesis. We suggest the authors to present this manuscript with more attractive hypothesis and to make the manuscript more interesting.

Response: Thank you very much for the comments. “The present study hypothesized that…and then possibly affects production performance” has been added in line 78, 80.

Conclusion

(again): Conclusion must be made based on the premises described at the final of introduction section. Conclusion strictly related to the results obtained. Depending on the assumptions/suggestions/probability/hypothesis, it can be considered in the discussion. Review the entire manuscript.

Response: Thank you very much for the comments. “Results showed that no significant differences found in the growth performance except for the product yield using different culture mode (PFI vs WF)…... For muscle composition,…While for the gut microbiota,…, corresponding with the higher rate of death, fatty liver disease, frequency and duration of cyanobacteria outbreaks…” has been added in the revised version in line 22-23, 24, 26, 33-34, etc.

Figures: poor quality!

Response: Thank you very much for the comments. The authors provided and uploaded (as attachments) the high resolution of figures in the revised version. The previous poor quality mainly attributed for the merged figure format.

Reviewer 2 Report

The study seems to be suitable for publication after some minor modifications:

Title should be modified into " the effects of different feeding regimes on body composition, gut microbial population, and susceptibility to pathogenic infection in largemouth bass"

The abstract section poorly rewritten. The authors missed to detailed treated groups, how many fish in each group, and how many replicate, what type of ponds applied? The experimental period, and the initial fish weight? The mentioned results not represented well and the final conclusion not cleared?

MS-222 dose should be addressed?

The authors should identify which part of intestine had been used (anterior, posterior or mid part)?

The authors should be aware about the journal guidelines in inserting the references in the text and references list? And line numbering should be emended?

The introduction and discussion part seems good but need minor style and English edited.

The conclusion part needs to be more concise and informative.

the English editing need minor revision

Author Response

The study seems to be suitable for publication after some minor modifications:

Title should be modified into "the effects of different feeding regimes on body composition, gut microbial population, and susceptibility to pathogenic infection in largemouth bass"

Response: Thank you, corrected as suggested in line 2-4.

The abstract section poorly rewritten. The authors missed to detailed treated groups, how many fish in each group, and how many replicate, what type of ponds applied? The experimental period, and the initial fish weight? The mentioned results not represented well and the final conclusion not cleared?

Response: Thank you very much for the comments. The abstract has been revised as “This study investigated the effects of dietary commercial feed (n=50025 in triplicate, named group PF for soil dike pond, sampling n=7; n=15000 in triplicate, WF for water tank, n=8), iced fish (n=50025 in triplicate, PI, n=7) and a combination of both (n=50025 in triplicate, PFI, n=8), on different metabolic parameters of the largemouth bass, Micropterus salmoides (0.67±0.09 g, culture period from June 2017 to July 2018). Throughout the experimental period, different areas of water (including input water of the front, middle of the pond and from the drain off at the back) and their mixed samples were simultaneously analyzed to find the source of the main infectious bacteria. Various feeding strategies may differentially affect body composition and shape the gut microbiota, but the mode of action has not been determined. Results showed that no significant differences were found in the growth performance except for the product yield using different culture mode (PFI vs WF). For muscle composition, the higher ∑SFA, ∑MUFA, ∑n-6PUFA, and 18:3n-3/18:2n-6 levels were detected in largemouth bass fed with the iced fish, while enrichment in ∑n-3PUFA and ∑HUFA was detected in largemouth bass fed with commercial feed. While for the gut microbiota, Fusobacteria, Proteobacteria, and Firmicutes were the most dominant phyla among all the gut samples. The abundance of Firmicutes and Tenericutes significantly decreased and later increased with iced fish feeding. The relative abundance of species from the Clostridia, Mollicutes, Mycoplasmatales, families (Clostridiaceae and Mycoplasmataceae) significantly increased in feed plus iced fish (PFI) group relative to that in iced fish (PI) group. Pathways of carbohydrate metabolism and the digestive system were enriched in the commercial feed group, whereas infectious bacterial disease resistance related pathways were enriched in iced fish group, corresponding with the higher rate of death, fatty liver disease, frequency and duration of cyanobacteria outbreaks. Feeding with iced fish resulted in more activities in the digestive system and energy metabolism, more efficient fatty acid metabolism and had higher ∑MUFA, and simultaneously has the potential protection against infectious bacteria from the environment through change of intestinal microbiota in the pond of largemouth bass culturing. Finally, the difference in feed related to the digestive system may contribute to the significant microbiota branch in the fish gut, and the input and outflow of water affects the intestinal flora in the surrounding water and in the gut, which in turn affects growth and disease resistance.”.

MS-222 dose should be addressed?

Response: Thank you, the dosage “150 mg/L,” has been added in line 125.

The authors should identify which part of intestine had been used (anterior, posterior or mid part)?

Response: Thank you, we identified the “foregut” as suggested, see line 146-7.

The authors should be aware about the journal guidelines in inserting the references in the text and references list? And line numbering should be emended?

Response: Thank you very much for the comments. The reference inserting and its list followed by the journal guidelines, and the line numbering has been added.

The introduction and discussion part seems good but need minor style and English edited.

Response: Thank you very much for the comments. The revised version (i.e. line 18-20 abstract section, 50 introduction section, 92 Materials and Methods section, 281 Results section, 573 discussion section, etc.) has been polished by native English speaker Brian (editage Co., Ltd., China) and Ampeire Yona, the style has also been revised.

The conclusion part needs to be more concise and informative.

Response: Thank you very much for the comments. The sentence “These changes were not observed in the gut samples, revealing the importance of further studying the effects of different feeding strategies or habitats in the gut, of which data collated will help stimulate the development of LB culture.” has been deleted. “Results showed no significant differences found in the growth performance except for the product yield using different culture mode (PFI vs WF)…...While for the gut microbiota,…However, the higher rate of death, fatty liver disease, frequency and duration of cyanobacteria outbreaks increased in PI…Feeding with iced fish resulted in more activities in the digestive system and energy metabolism, more efficient fatty acid metabolism for owning the higher ∑MUFA, and simultaneously has the potential protection for infectious bacteria from the environment through the change for intestinal microbiota in the pond of largemouth bass culturing. Finally, the difference in feed related to the digestive system may contribute to the significant microbiota branch in the fish gut, and the input and outflow of water affect the intestinal flora in the surrounding water and in the gut, which in turn affects growth and disease resistance.” has been added in the revised version in line 684-5, 688, 692-3, 696-703.

the English editing need minor revision

Response: Thank you very much for the comments. The revised version has been polished by native English speaker Brian (editage Co., Ltd., China) and Ampeire Yona, and checked by all the authors.

Reviewer 3 Report

The authors present a study titled, " Different feeding strategies shape body composition, gut microbiota, and susceptibility to pathogenic infection in largemouth bass." While the article is well-structured and demonstrates significant effort, some areas require improvement and clarification:

  1. The abstract's opening sentences need revision. The second sentence should be placed before the first sentence to improve the flow.
  2. Figure 1 should either be divided into two separate figures or relocated to the results section, as its current placement in the Method section is not appropriate.
  3. The overall quality of figures 2 to 5 is poor, with small letters that are difficult to read and interpret. Please enhance the quality of these figures to facilitate better understanding.
  4. In the discussion section, there are several typographical errors, such as "heavy mental" and "microbiata" on page 15. Please correct these errors for clarity and accuracy.
  5. Concerning the relationship between water quality and gut microbiome in fish, is it solely related to dissolved oxygen levels, or are there other factors involved?
  6. In conclusion, explain the results’ relevance and potential applications in aquaculture. 

The article is clearly written and comprehensible, making it accessible to readers. As a non-native speaker, I may not be fully qualified to evaluate the language with absolute certainty. Still, overall, the text appears to be of high quality and easy to follow.

Author Response

The authors present a study titled, "Different feeding strategies shape body composition, gut microbiota, and susceptibility to pathogenic infection in largemouth bass." While the article is well-structured and demonstrates significant effort, some areas require improvement and clarification:

The abstract's opening sentences need revision. The second sentence should be placed before the first sentence to improve the flow.

Response: Thank you, corrected as suggested in line 12-14, 21-22.

  1. Figure 1 should either be divided into two separate figures or relocated to the results section, as its current placement in the Method section is not appropriate.

Response: Thank you, corrected as suggested in line 105-7.

  1. The overall quality of figures 2 to 5 is poor, with small letters that are difficult to read and interpret. Please enhance the quality of these figures to facilitate better understanding.

Response: Thank you, corrected as suggested. The figures separated into different parts to enhance the readability in the revised version in line 355-513.

  1. In the discussion section, there are several typographical errors, such as "heavy mental" and "microbiata" on page 15. Please correct these errors for clarity and accuracy.

Response: Thank you very much for the comments, corrected as suggested in line 567, 572.

  1. Concerning the relationship between water quality and gut microbiome in fish, is it solely related to dissolved oxygen levels, or are there other factors involved?

Response: Thank you very much for the comments. Gut microbiome were correlated with multiple factors, we revised related contents as follows.

“Environmental factors like water quality (DO, some toxicant like heavy metal) and treatment composition (like zooplankton enhanced Cetobacterium and Rhizobium, phytoplankton, etc) [5,10,15], nutritional enzyme activity, and disease may shape gut microbiota [16].” in line 567-71, and only “higher DO, with lower SD/TSS/TN/TP…higher product yield” in line 638-9 detected in the current study, which showed “a consistent association was observed between β-diversity of the gut microbiota and dissolved oxygen concentration in water when compared with PF [29]” in line 612-3.

  1. In conclusion, explain the results’ relevance and potential applications in aquaculture. 

Response: Thank you very much for the comments. “Feeding with iced fish resulted in more activities in the digestive system and energy metabolism, more efficient fatty acid metabolism for owning the higher ∑MUFA, and simultaneously has the potential protection for infectious bacteria from the environment through the change for intestinal microbiota in the pond of largemouth bass culturing. Finally, the difference in feed related to the digestive system may contribute to the significant microbiota branch in the fish gut, and the input and outflow of water affect the intestinal flora in the surrounding water and in the gut, which in turn affects growth and disease resistance.” has been added in the revised version in line 696-703. Concerning this query, “From the perspective of animal health farming, we may need to adjust feeding strategies, strengthen the control of water and drug intake/drainage, to adapt to the ice fresh fish replacement action proposed by the Ministry of Agriculture and Rural Affairs.” simultaneously be added in line 703-6.

The article is clearly written and comprehensible, making it accessible to readers. As a non-native speaker, I may not be fully qualified to evaluate the language with absolute certainty. Still, overall, the text appears to be of high quality and easy to follow.

Response: Thank you very much for your recognition of this Ms., we believe that with your contribution, the quality of this Ms. will be greatly improved.

Reviewer 4 Report

Pag. 2-3 Materials and Methods: please describe better the experimental condition (also with a scheme)

Fig.1 : divided in Sampling point and fatty acid results

3.1 Water quality and disease incident: please indicate the results (Table)

3.2 Growth performance: please indicate the results (Table)

3.3 Body composition: please indicate the results (Tab and Figure 1B)

Author Response

Pag. 2-3 Materials and Methods: please describe better the experimental condition (also with a scheme)

Response: Thank you very much for the comments. We separated Fig.1 into two panels, and added the scheme in the Fig.1 to make the potential readers clearly in line 105-7.

Fig.1 : divided in Sampling point and fatty acid results

Response: Thank you very much for the comments. Corrected as suggested in line 105-7, 317.

3.1 Water quality and disease incident: please indicate the results (Table)

Response: Thank you very much for the comments. We added the citation of Table S2 in the main text as suggested in the revised Ms in line 267, 269, 274, 276, 279.

3.2 Growth performance: please indicate the results (Table)

Response: Thank you very much for the comments. We added the citation of Table S1 in the main text as suggested in the revised Ms in line 285-6.

3.3 Body composition: please indicate the results (Tab and Figure 1B)

Response: Thank you very much for the comments. We added the citation of Figure 1B in the main text as suggested in the revised Ms in line 295, 299, 302, 309, 313, 316 for Figure 2 and line 179 for Table S1.

Round 2

Reviewer 4 Report

Please indicate in all captions the significate of the acronyms PF, WF, PI PFI

The Figure caption could be simplify and are not clear the letter indication (A, B) in the caption

Author Response

Please indicate in all captions the significate of the acronyms PF, WF, PI PFI

Response: Thank you very much for the comments. The significate of the acronyms have been added in all the captions (line 373, 407-8, 423-4, 574-5, 607-8, 619-20).

The Figure caption could be simplify and are not clear the letter indication (A, B) in the caption

Response: Thank you, and the Figure captions have been revised (line 102, 358-9).